# GRT-X Stimulates Dorsal Root Ganglia Axonal Growth in Culture via TSPO and Kv7.2/3 Potassium Channel Activation

**DOI:** 10.3390/ijms25137327

**Published:** 2024-07-03

**Authors:** Léa El Chemali, Suzan Boutary, Song Liu, Guo-Jun Liu, Ryan J. Middleton, Richard B. Banati, Gregor Bahrenberg, Rainer Rupprecht, Michael Schumacher, Liliane Massaad-Massade

**Affiliations:** 1Maladies et Hormones du Système Nerveux, Inserm, Université Paris-Saclay, 94276 Le Kremlin-Bicêtre, France; lea.el-chemali@inserm.fr (L.E.C.); suzan.boutary@inserm.fr (S.B.); song.liu@inserm.fr (S.L.); 2Australian Nuclear Science and Technology Organisation (ANSTO), Kirrawee, NSW 2232, Australia; gdl@ansto.gov.au (G.-J.L.); rym@ansto.gov.au (R.J.M.); 3Faculty of Medicine and Health, University of Sydney, Camperdown, NSW 2006, Australia; richard.banati@professoriate.org; 4Global Preclinical R&D, Grünenthal Innovation, Grünenthal GmbH, Zieglerstraße 6, D-52078 Aachen, Germany; gregor.bahrenberg@grunenthal.com; 5Department of Psychiatry and Psychotherapy, University of Regensburg, D-93053 Regensburg, Germany; rainer.rupprecht@medbo.de

**Keywords:** TSPO, peripheral benzodiazepine receptor, potassium channels, Kv7.2/3, KCNQ2/3, GRT-X, axonal growth, gene expression, dorsal root ganglia

## Abstract

GRT-X, which targets both the mitochondrial translocator protein (TSPO) and the Kv7.2/3 (KCNQ2/3) potassium channels, has been shown to efficiently promote recovery from cervical spine injury. In the present work, we investigate the role of GRT-X and its two targets in the axonal growth of dorsal root ganglion (DRG) neurons. Neurite outgrowth was quantified in DRG explant cultures prepared from wild-type C57BL6/J and TSPO-KO mice. TSPO was pharmacologically targeted with the agonist XBD173 and the Kv7 channels with the activator ICA-27243 and the inhibitor XE991. GRT-X efficiently stimulated DRG axonal growth at 4 and 8 days after its single administration. XBD173 also promoted axonal elongation, but only after 8 days and its repeated administration. In contrast, both ICA27243 and XE991 tended to decrease axonal elongation. In dissociated DRG neuron/Schwann cell co-cultures, GRT-X upregulated the expression of genes associated with axonal growth and myelination. In the TSPO-KO DRG cultures, the stimulatory effect of GRT-X on axonal growth was completely lost. However, GRT-X and XBD173 activated neuronal and Schwann cell gene expression after TSPO knockout, indicating the presence of additional targets warranting further investigation. These findings uncover a key role of the dual mode of action of GRT-X in the axonal elongation of DRG neurons.

## 1. Introduction

Adult peripheral nerves retain the ability to regenerate after a lesion thanks to the intrinsic capacity of neurons to regrow their axons, the permissive environment provided by Schwann cells, which transdifferentiate into repair cells, and infiltrating macrophages [1,2]. In response to axonal injury, neurons change from a transmitter to a pro-regenerative state, with the upregulation of genes involved in signaling pathways supporting their survival and neurite outgrowth [2,3]. Schwann cells adopt a repair phenotype by downregulating myelin-associated genes and by upregulating genes supporting repair and regeneration, coding for neurotrophic factors and cytokines [4], and forming bands of Büngner that provide structural scaffolds for the regenerating axons [5]. However, both the intrinsic ability of axons to regrow and the regenerative support provided by Schwann cells decrease with time and lead to insufficient functional recovery [6].

Insufficient peripheral nerve regeneration and functional restoration may also lead to the development of chronic neuropathic pain (CNP) [7,8], and no adequate treatment is available [9]. Stimulating and preserving the intrinsic capacity of axonal growth over time as well as preventing the chronification of neuropathic pain by pharmacological interventions thus represents an important therapeutic objective for the treatment of traumatic injuries.

Opioids are generally prescribed to treat severe pain. However, they are not very effective, do not prevent the development of CNP from acute pain, and may result in harmful side effects linked to the development of opioid tolerance and dependence [10]. This highlights the need to find new treatments for neuropathic pain and to prevent its chronification.

Slow voltage-gated Kv7 K^+^ channels have been identified as attractive drug targets for nerve injury pain as they regulate neuronal excitability and are strongly linked to CNP [11,12]. These channels elicit slow-activating, non-inactivating, and voltage-dependent K^+^ currents (M-currents) that exert an inhibitory control on neuronal excitability [13]. Their opening has been shown to inhibit the firing of action potentials in DRG sensory neurons and to inhibit nociception [14]. Moreover, the expression of Kv7 channels in DRG neurons is up- or downregulated following axotomy, depending on the subtype [15,16].

Of the five M-type Kv7 K^+^ channels (Kv7.1–Kv7.5) encoded by the KCNQ genes, Kv7.2/3 are the major regulators of neuronal excitability [17]. First developed as an anti-epileptic drug, retigabine opens the Kv7.2-5 channels, induces a hyperpolarizing shift, and is effective and neuroprotective for the treatment of neuropathic pain in preclinical rodent models [11,18]. This anticonvulsant was withdrawn from the market in 2017 due to safety issues and limited use.

Following the hypothesis that Kv7.2/3-selective channel openers will be safer and more efficient, efforts are underway to develop more subtype-selective compounds [19]. The Kv7.2/3 channel-preferring channel opener ICA27243, developed as an anti-epileptic drug, has been a step forward in this direction [20]. Later on, GRT-X (N-[(3-fluorophenyl)-methyl]-1-(2-methoxyethyl)-4-methyl-2-oxo-(7-trifluoromethyl)-1H-quinoline-3-caboxylic acid amide) was described as a selective Kv7.2/3 activator [21]. In addition, GRT-X was shown to be a potent activator of the rat mitochondrial translocator protein 18 kDa (TSPO), formerly known as the peripheral benzodiazepine receptor [22,23]. This mechanism of dual activation of Kv7.2/7.3 and TSPO by GRT-X was proposed to explain that the compound is anti-hyperalgesic in rodents, promotes the viability of sensory and motor neurons, and accelerates peripheral nerve regeneration and functional recovery after a severe crush lesion of rat cervical spinal nerves [22]. The activation of TSPO is known to be associated with regenerative responses in neuronal injury and inflammatory processes [24,25,26,27]. More recently, a role of TSPO in neuronal plasticity has been demonstrated in global TSPO knockout mice [28]. The activation of Kv7.2/7.3 has also been shown to protect neurons against injury and may contribute to peripheral nerve regeneration by reducing injury-induced hyperexcitability of DRG neurons [29,30,31].

As reviewed elsewhere, TSPO is a fundamental regulator of physiological functions and cellular processes, such as apoptosis, proliferation, and the synthesis of steroids [26,32,33]. Several TSPO ligands appear to promote neuroprotection, alleviate pain, increase neuronal survival, accelerate axonal regrowth, and improve functional recovery in rodent lesion models [34,35,36]. However, the verification and precise role of TSPO in these experimental observations could not be clarified because of the absence of pharmacological TSPO antagonists with confirmed specificity and selectivity. The availability of TSPO-KO mice that are fully viable and have intact vital functions [37,38] now allows for more unequivocal confirmation of TSPO drug actions.

Here, we assessed the role of GRT-X on neurite growth in embryonic E13.5 dorsal root ganglion (DRG) explant cultures prepared from C57BL6/J or TSPO-KO mice [37]. We compared its effects to those of other treatments, including the TSPO ligand XBD173 (AC-5216, Emapunil), the Kv7.2/3 agonist ICA27243, and the Kv7.2/3 antagonist XE991. We also evaluated and compared the effects of the different treatments on the expression levels of genes coding for the so-far-identified GRT-X targets (Kv7.2, Kv7.3, and TSPO); genes involved in the timing of Schwann cell developmental transitions including cadherin 19 (Cad19, Cdh19); transcription factors AP-2alpha (Tfap2α, AP-2) and Krox20; genes implicated in myelin formation or coding for major myelin structural proteins comprising desert hedgehog (Dhh), the 2′,3′-cyclic nucleotide 3′-phosphodiesterase (CNPase), myelin protein zero (Mpz), myelin basic protein (Mbp), and myelin proteolipid protein (Plp); as well as genes involved in axonal maintenance and outgrowth, in particular stathmin-2 (Stmn2), peripherin, and neurofilament heavy (Nfh). Based on previous published work performed by Sundaram V.K. et al., all these genes are shown to be expressed in E13.5 mouse DRG neuron/Schwann cell cultures [39].

## 2. Results

### 2.1. GRT-X Promotes Neurite Growth in Embryonic C57BL6/J DRG at DIV4

We conducted an initial assessment of the positive impact of 10 µM GRT-X on neurite network growth in E13.5 mouse DRG explants at DIV4 (Appendix A). Subsequently, we compared the effects of GRT-X on axonal growth at DIV4 to those of one TSPO ligand (XBD173 at 10 µM), a Kv7.2/3 antagonist (XE991 at 10 µM), a Kv7.2/3 agonist (ICA27243 at 10 µM), and the control vehicle (0.1% ethanol). DRG explants were obtained from embryos at E13.5, treated at DIV1, and cultured until DIV4, followed by fixation with 4% PFA (Figure 1A). Axons were immunolabeled with anti-NFH, and cell nuclei were stained with DAPI (Figure 1B). Each group comprised three to four DRGs, and the experiment was repeated four times. The number of intersections at varying distances (µm) (Figure 1C) and the areas under the curve (AUCs) (Figure 1D) were calculated.

The results confirmed that GRT-X significantly enhances the length and density of the neurite network at DIV4 (see Figure 1B, second left column), with a marked increase in AUC compared to the other groups. In contrast, the XBD173, XE991, ICA27243, and EtOH treatments exhibited no significant effect on neurite growth.

### 2.2. GRT-X Maintains Its Positive Effect on Axonal Growth in E13.5 DRG Explants at DIV8

To assess the durability of GRT-X’s positive effects over time, DRG explants from E13.5 embryos were either not treated (NT) or treated at DIV1 with a vehicle (0.1% EtOH), GRT-X, XBD173, XE991, or ICA-27243 (all at 10 µM), and the cultures were terminated at DIV8 and subsequently fixed in 4% PFA (Figure 2A–D). The results show that the AUC of the GRT-X-treated group was significantly higher than that of the other groups. This suggests that the positive effect of GRT-X is sustained over time up to DIV8, and that a single treatment dose is adequate to promote axonal growth for at least one week in vitro.

### 2.3. GRT-X and XBD173 Increase Axonal Growth in E13.5 DRG Explants at DIV8 after Two Treatments

The objective of this experiment was to assess the impact of various treatments, repeated twice, on axonal growth. DRG explants from E13.5 embryos were initially treated at DIV1 and subsequently at DIV4 with a vehicle (0.1% EtOH), GRT-X, XBD173, XE991, or ICA-27243 (all at 10 µM), and the cultures were terminated at DIV8 and subsequently fixed in 4% PFA (Figure 3A–D). Again, one group received no treatment to rule out any vehicle effects. The AUCs for the GRT-X- and XBD173-treated groups were significantly higher when compared to the other treatment groups.

### 2.4. GRT-X Induces Increases in the Expression of Genes Involved in Myelination, Schwann Cell and Neuronal Development and Differentiation, and Axonal Structure

Upon GRT-X incubation, the expression changes of genes associated with myelination, Schwann cell differentiation, and axonal structure were analyzed and compared with those induced by XBD173, XE991, and ICA-27243. Dissociated cultures of E13.5 DRGs were treated at DIV1 in the following groups: GRT-X, XBD173, XE991, ICA-27243 (all at 10 µM), and EtOH 0.1%. Again, one group received no treatment (NT). At DIV4 of culture, total RNA was extracted separately from the respective wells for RT-qPCR studies (Figure 4). The mRNA levels of TSPO, Kv7.2, Kv7.3, Mpz, Mbp, Plp, Dhh, Krox20, CNPase, Tfap2α, Peripherin, Cad19, and Nfh were quantified relative to Gapdh mRNA levels (Figure 4A). Each group consisted of 37 to 40 DRGs, and the experiment was repeated three times.

All treatments induced a significant increase in TSPO and Kv7.2 expression, with the GRT-X and XBD173 treatments leading to significantly higher increases compared to the other groups (Figure 4B). However, only the GRT-X and XBD173 treatments induced a significant increase in Kv7.3 mRNA levels. Regarding genes involved in myelination, both the GRT-X and XBD173 treatments promoted increased total mRNA expression levels of myelin-related genes (Mpz, Mbp, and Plp), with GRT-X having a significantly greater effect (Figure 4C). On the other hand, XE991 and ICA27243 decreased mRNA expression levels of the Mpz and Mbp genes but had no effect on Plp. For genes related to Schwann cell development and differentiation, the GRT-X and XBD173 treatments significantly increased the relative expression of the Dhh, Krox20, and CNPase genes, with XBD173 having a significantly greater impact on Dhh expression and GRT-X on Krox20 and CNPase (Figure 4D). XE991 had no effect on Dhh gene expression but increased Krox20 expression and decreased CNPase expression. ICA-27243 had no effect on Dhh expression but led to a decrease in Krox20 and CNPase mRNA expression.

Regarding genes associated with neuronal development and differentiation, GRT-X and XBD173 increased total mRNA expression levels of the genes Tfap2α, Stmn2, Peripherin, and Cad19, with GRT-X having a significantly higher effect (except for Tfap2α) (Figure 4E). XE991 had no effect on gene expression, while ICA-27243 decreased the mRNA expression levels of all the listed genes. GRT-X and XBD173 increased Nfh mRNA levels, with GRT-X having a significantly higher effect (Figure 4F). XE991 and ICA-27243 reduced Nfh mRNA levels. For all the genes studied, no differences were found between the NT and the ethanol-treated groups. At the low concentration used (0.1%), ethanol does not affect neurite outgrowth.

### 2.5. GRT-X Has No Effect on Neurite Growth in Embryonic TSPO-KO DRG Explants at DIV4

To validate that the impact of GRT-X on axonal growth is mediated by TSPO, we conducted experiments using embryonic DRG explants obtained from TSPO-KO embryos at E13.5. These TSPO-KO explants were selected as tools since selective pharmacological inhibitors of TSPO are lacking. All other procedures were as in the WT mice and are described in Figure 1.

Our findings demonstrate that GRT-X and XBD-173 do not influence axonal growth after knockout of TSPO expression, confirming that both compounds exert their axonal-promoting effects via TSPO. Correspondingly, the areas under the curve (AUCs) for the groups treated with XE991 and ICA27243 were significantly lower than those for all the other groups (Figure 5A–D).

### 2.6. GRT-X Increases the Expression of Genes Related to Myelination and Schwann Cell and Neuron Development, but Not Those Involved in Axonal Structure in Dissociated Cultures of Embryonic TSPO-KO DRGs

To validate the confocal microscopy results, we established dissociated cultures from embryonic DRG explants obtained from TSPO-KO embryos at E13.5. These cultures were treated again at DIV1 with GRT-X, XBD173, XE991, ICA-27243, or a vehicle (EtOH 0.1%), and one group received no treatment (NT). At DIV4, total RNA was extracted from the samples for RT-qPCR studies. The mRNA levels of Kv7.2, Kv7.3, Mpz, Mbp, Plp, Dhh, Krox20, CNPase, Tfap2α, Peripherin, Cad19, and Nfh were quantified relative to Gapdh mRNA levels (Figure 6A). Each group consisted of 37 to 40 DRGs per group, and the experiment was repeated three times.

In the TSPO-KO mice, our findings indicate that GRT-X, XBD173, and XE991 led to an increase in Kv7.2 mRNA levels, with GRT-X having a significantly stronger effect (Figure 6B). Conversely, ICA-27243 resulted in a reduction in relative Kv7.2 expression. Moreover, GRT-X promoted an increase in Kv7.3 mRNA levels, while XE991 and ICA-27243 induced a significant decrease (Figure 6B).

Concerning genes associated with myelination, GRT-X enhanced the mRNA levels of all genes (Mpz, Mbp, and Plp). XBD173 and XE991 also increased Mbp expression, though the impact of GRT-X remained significantly stronger. Conversely, Plp expression was reduced by XE991 and ICA-27243 (Figure 6C).

For the genes related to Schwann cell development and differentiation, our findings reveal that GRT-X enhanced the mRNA levels of the Dhh and CNPase genes but did not affect Krox20 expression. In contrast, XBD173 increased the expression of all three genes (Dhh, Krox20, and CNPase). Conversely, XE991 reduced the mRNA levels of Krox20 and CNPase. ICA-27243 decreased Krox20 expression but did not influence the other two genes (Figure 6D).

In terms of genes associated with neuronal development and differentiation, GRT-X increased Tfap2α and Cad19 gene expression while decreasing Stmn2 expression, with no impact on peripherin. On the other hand, XBD173 elevated Tfap2α and peripherin mRNA levels but had no effect on Stmn2 and Cad19 gene expression. XE991 led to an increase in Tfap2α expression and a decrease in Stmn2 expression. ICA-27243 caused a reduction in Stmn2 mRNA levels (Figure 6E). Regarding the axonal Nfh gene, the ICA-27243 treatment resulted in reduced mRNA levels. Conversely, the other treatments had no effect (Figure 6F). Notably, ethanol had no impact on the expression of the aforementioned genes. This is due to the low concentration of ethanol (0.1%) used in our study.

## 3. Discussion

In this study, the influence of GRT-X on the axonal growth of rodent DRG neurons was explored, and the contribution of the molecular targets TSPO and Kv7.2/3 was analyzed. For this purpose, both explant and dissociated cultures of embryonic DRG derived from C57BL6/J or TSPO-KO mice, respectively, were used at E13.5. These cultures derived from embryonic DRGs allow us to study the role of GRT-X and TSPO in early axonal and Schwann cell development [39]. Of note, the vast majority of the DRG neurons still express trkA at this early developmental stage and survive after supplementation with NGF, as is the case in our culture conditions [40].

Ganglionic and dissociated DRG cultures represent excellent models for investigating the molecular and cellular mechanisms underlying the role of GRT-X and TSPO in axonal elongation and Schwann cell plasticity under controlled conditions, with possible implications for axonal regeneration after injury or peripheral neuropathies [41,42]. Both adult peripheral neurons and Schwann cells respond to injury by reprogramming their phenotypes toward pro-regenerative conditions. However, although Schwann cells re-express developmental genes, one has to be aware of the differences in molecular mechanisms between repair and immature Schwann cells [43,44].

To analyze the molecular mechanisms involved in developmental and repair processes, GRT-X, which is supposed to exert neuroregenerative effects [22], was tested. Only GRT-X significantly increased neurite length and density compared to the TSPO ligand (XBD173) and Kv7.2/3 modulators (ICA-27243 and XE991) (refer to Appendix A for an overview).

Interestingly, both XE991 and ICA27243 showed a trend to decrease the AUC. Indeed, at the early embryonic stage (E13.5), mouse DRG neurons are still undergoing significant structural and functional maturation, and they exhibit particular electrophysiological properties [45]. Their modulation by activating or inhibiting Kv7.2/3 channels may account for the observed experimental outcomes.

At DIV8, GRT-X maintains a positive axonal growth effect after a single treatment, while the other treatments show no impact. However, following two successive treatments, XBD173 also stimulated axonal elongation and increased the AUCs, with the effects of GRT-X and XBD173 surpassing those of the other treatments. Thus, both the dual mode of action of GRT-X and the supposedly selective TSPO activator XBD173 can promote axonal growth.

To strengthen TSPO involvement in axonal growth, the knockout of TSPO mice resulted in the complete loss of the stimulatory effects of GRT-X on axon growth, demonstrating TSPO importance in culture at the early embryonic stage. The effects of TSPO ligands on neuronal differentiation have so far mainly been demonstrated in neuronal cell lines [46]. The non-selective TSPO ligand 4′-chlorodiazepam, which is also active at GABA_A_ receptors, has previously been shown to increase neurite formation in primary cultures of mouse hippocampal neurons at E17 [47]. TSPO expression in the nervous system has been demonstrated in situ as early as E12 [48]. Of note, in TSPO knockout DRGs, the Kv7.2/3 ligands ICA27243 and XE991 significantly inhibited axonal elongation. Although Kv7 activation protects neurons against injury-induced hyperexcitability [29,30], it has been shown to inhibit neurite growth in cultured neuronal cells [49]. Thus, the double target action of GRT-X may compensate for the negative influence of Kv7 channel activation on axonal elongation.

In rodent cultures, GRT-X was more effective than XBD173 in stimulating axonal growth via TSPO, showing significant effects 4 days after a single dose, while XBD173 only showed effects at DIV8 after two doses. The different capabilities of both compounds to stimulate axonal growth cannot be explained by their apparent affinities for TSPO, as they both show estimated Ki values in the nanomolar range [22,50]. The efficacy of TSPO ligands is difficult to predict from their affinities and pharmacokinetics, possibly because TSPO is part of a mitochondrial multi-protein complex [32,51,52,53]. Moreover, TSPO binding sites for different ligands remain to be characterized, as do the 3D structures of TSPO complexed with different ligands at different sites [54].

Moreover, GRT-X increased the expression levels of all the selected genes in C57BL/6 DRG cultures (summary in Table 1). These findings are consistent with our confocal microscopy results, demonstrating that GRT-X increases the length and density of Nfh-labeled axons. In addition, the increase in the expression of myelin-related genes shows that GRT-X modulates mechanisms that depend on both neurons and Schwann cells, which are essential for the development of peripheral nerves. Importantly, all the studied neuronal and Schwann cell genes play important roles in both peripheral nerve development and adult nerve regeneration [55,56,57,58,59,60,61,62]. Thus, our findings are likely to be relevant for both developmental and regenerative processes, as we are allowed to conclude from our ex vivo setup. GRT-X has indeed been shown to promote the survival and regeneration of sensory and motor neurons and to accelerate the recovery of sensory functions after a severe crush lesion of cervical spinal nerves in adult rats [22].

Notably, treatment with XBD173 also increased the expression of all the genes, contrasting with its lack of effect on axonal elongation in the ganglionic DRG cultures after a single administration. In contrast, the Kv7.2/3 ligands ICA-27243 and XE991 inhibited the expression of most of the genes, suggesting that Kv7.2/3 activation does not contribute to the greater efficacy of GRT-X when compared to XBD173.

In DRG explant cultures from TSPO-KO embryos, GRT-X lost its ability to enhance axonal growth. However, it still stimulated most gene expressions in TSPO-KO, except for the neuronal genes peripherin and Nfh (axonal outgrowth) and Krox20 (Schwann cell maturation). It is unlikely that Kv7 activation contributed to the upregulation of the 10 genes by GRT-X in the absence of TSPO, as ICA-27243 had either no effect or decreased their expression. A more likely explanation would be the potential presence of a third or other multiple targets of GRT-X, warranting further molecular investigations for confirmation. Surprisingly, even the supposedly selective TSPO activator XBD173 increased the expression of the key neuronal and Schwann cell genes Kv7.2, Mpz, Dhh, Krox20, CNPase, Tfap2α, and peripherin in the TSPO-KO DRG. This strongly suggests that XBD173 may activate these genes by acting through other targets than TSPO. From these observations, it follows that pharmacological tools may not always be sufficient to delineate the role of a target, but that expression studies provide additional important information. Consistent with the inhibitory effects of ICA-27243 and XE991 on axonal elongation, both compounds decreased or had no effect on the expression of most of the genes. The only exception was the increase in Kv7.2, Mpz, and Tfap2α by XE991, which may oppose the inhibitory effects of intrinsic Kv7.2/3 activity.

Our data demonstrate that the potent stimulatory effects of GRT-X on axonal growth require activation of its TSPO target. Interestingly, whereas neuron hyperpolarization by Kv7.2/3 activation has been reported to inhibit neurite growth, which is consistent with the downregulation of neuronal and Schwann cell genes observed here, dual activation of the potassium channels and TSPO by GRT-X results in a strong axon elongation effect. TSPO activation by GRT-X may thus compensate for Kv7.2/3-mediated inhibition in vitro. Indeed, the Kv7.2/3 ligands ICA-27243 and XE991 significantly inhibited axonal growth in TSPO-KO mice. However, activation of TSPO alone by XBD173 was much less efficient in stimulating axon elongation than the targeting of both TSPO and Kv7.2/3 channels by GRT-X. These results further support the concept of a novel dual mode of action for GRT-X that would be potentially additive or synergistic, dual mode of action for GRT-X [22]. However, the underlying mechanism remains to be uncovered. The intriguing observation that both GRT-X and XBD173 activate the expression of neuron and Schwann cell genes in TSPO-KO mice strongly suggests the existence of additional targets for both compounds, which may also contribute to the strong effects of GRT-X.

Thus, GRT-X could have an interesting therapeutic potential by combining neuropathy pain relief via the Kv7.2/3 channels with neuroregenerative benefits via TSPO, warranting further translational studies. It is, however, important to note that significant differences are observed between neuronal populations during the embryonic and mature stages. At E13.5, approximately 80% of neurons within mouse embryos are TrkA-positive and engage in active proliferation and neurite extension. Conversely, in adult mice, this proportion drops to approximately 40%, in which neurons are mature and have acquired distinct sensory functionalities [63]. These distinctions necessitate consideration when investigating the in vivo effects of GRT-X.

**Table 1 ijms-25-07327-t001:** Comparison of treatment effects on gene expression in DIV4 dissociated DRG cultures from E13.5 pregnant C57BL/6 and TSPO-KO mice.

	Embryonic C57BL/6 DRGs					Embryonic TSPO-KO DRGs				
	GRT-X	XBD173	XE991	ICA-27243	EtOH	GRT-X	XBD173	XE991	ICA-27243	EtOH
TSPO	↑	↑	↑	↑	-	N/A	N/A	N/A	N/A	-
Kv7.2	↑	↑	↑	↑	-	↑	↑	↑	↓	-
Kv7.3	↑	↑	-	-	-	↑	-	↓	↓	-
Mpz	↑	↑	↓	↓	-	↑	↑	↑	-	-
Mbp	↑	↑	↓	↓	-	↑	-	-	-	-
Plp	↑	↑	-	-	-	↑	-	↓	↓	-
Dhh	↑	↑	-	-	-	↑	↑	-	-	-
Krox20	↑	↑	↑	↓	-	-	↑	↓	↓	-
CNPase	↑	↑	↓	↓	-	↑	↑	↓	-	-
Tfap2α	↑	↑	-	↓	-	↑	↑	↑	-	-
Stmn2	↑	↑	-	↓	-	↑	-	↓	↓	-
Peripherin	↑	↑	-	↓	-	-	↑	-	-	-
Cad19	↑	↑	-	↓	-	↑	-	-	-	-
Nfh	↑	↑	↓	↓	-	-	-	-	↓	-

↑: increased gene expression; ↓: decreased gene expression; -: no effect on gene expression; N/A: not applicable for TSPO-KO DRGs because the gene is not expressed.

In conclusion, GRT-X exerts its effects in isolated rodent DRG neurons through dual targeting of TSPO and Kv7.2/3, resulting in the stimulation of axonal growth and upregulation of various gene expressions related to myelination, Schwann cell differentiation, and neuronal structure (Figure 1 summarizes our findings). The simultaneous activation of TSPO and Kv7.2/3 by GRT-X suggests its potential as a therapeutic candidate for nerve lesions and neuropathic pain. Future studies should explore combined treatments and in vivo nerve lesions to further validate these effects on axonal growth. 

The anti-epileptic and anti-hyperalgesic effects of GRT-X described by Bloms-Funke et al. [22] are putatively supported by the stimulation of axonal growth and modulation of the genes involved in myelination, Schwann cell differentiation, and neuronal structure.

## 4. Materials and Methods

### 4.1. Chemicals

GRT-X (N-[(3-fluorophenyl)-methyl]-1-(2-methoxyethyl)-4-methyl2-oxo-(7-trifluoromethyl)-1H-quinoline-3-caboxylic acid amide, chemical structure represented in Figure 7) was provided by Grünenthal GmbH (Aachen, Germany). XBD173 and XE991 were purchased from Sigma Aldrich Chimie (Saint-Quentin-Fallavier, France) (SML1223 and X2254, respectively). ICA27243 was purchased from TargetMol,T15545 (36 Washington Street, Wellesley Hills, MA 02481, USA). All compounds were dissolved in ethanol (final concentration of 0.1%) with a final concentration of 10 µM.

### 4.2. Animals and Tissue Harvesting

Our experiments were performed on timed-pregnant C57Bl6/J mice (Janvier Labs, Le Genest-Saint-Isle, France) and TSPO knockout (TSPO-KO) mice. Mice were housed in standard plastic cages in a 12 h/12 h light/dark cycle in a temperature-controlled room (21 °C), with ad libitum access to food and water. The care and use of mice were conformed to institutional policies and guidelines (INSERM, French and European Community Council Directive 86/806/EEC). The TSPO-KO mice were back-crossed with C57Bl6/J mice to maintain the genetic background of the original wild-type strain [37,38]. The pregnant mice were sacrificed by cervical dislocation after being anesthetized with isoflurane. Embryos were surgically removed and placed in ice-cold L-15 media (Gibco 11415-064, Grand Island, NY, USA). Spinal cords were extracted from these embryos, and DRGs were harvested based on established protocols [64,65].

### 4.3. DRG Explant Cultures

Following DRG harvesting as described above, one DRG explant was placed and seeded in the center of a circular plastic coverslip (LFG distribution 174950) (Figure 8A) previously coated with poly-L-lysine (PLL) (Sigma Aldrich Chimie Saint-Quentin-Fallavier, France) and collagen (R&D systems 3440-100-01; Minneapolis, MN, USA) (refer to Appendix A for coating details). The coverslips were placed on 4-well culture plates. The cells were first cultured in DRG Plating Media (refer to Appendix A for media composition) for 16 h to allow them to attach. The next day, the media were replaced by Neurobasal Media (refer to Appendix A for media composition) containing different treatments (NT, GRT-X 10 µM, XBD173 10 µM, XE991 10 µM, ICA2743 10 µM, EtOH 0.1%), and the cells were kept in culture for a certain period dependent on the groups (Table 2). For each experiment, *n* = 3–4 DRG explants were cultured per group, and in each group, every DRG originated from a different embryo. Each experiment was repeated 2–3 times.

### 4.4. Dissociated DRG Cultures

DRGs were extracted from C57BL6/J embryos as previously described. A total of 40 DRGs were harvested per embryo for each group (Figure 8B). Based on the experimental protocol developed by Sundaram et al. [39], DRGs were incubated in trypsin (0.25% trypsin in 1X HBSS) at 37 °C for 30 min. Trypsinization was stopped by adding L-15 medium containing 10% horse serum (Gibco, 26050088). The individual DRG solutions were centrifuged at 1500 rpm for 5 min. The supernatant was removed, and the tissues were resuspended in DRG Plating Media and triturated 30 times with a sterilized Pasteur pipette until a homogeneous cell suspension was obtained. For each treatment (NT, GRT-X, XBD173, XE991, ICA27243, EtOH), 40 dissociated DRGs were plated on 35 mm culture dishes each, which were previously coated with PLL (Sigma P7890) and collagen (R&D systems 3440-100-01). The cells were then cultured in DRG Plating Media for 16 h to allow the cells to attach. The next day, the culture media were replaced by Neurobasal Media containing different treatments (NT, GRT-X 10 µM, XBD173 10 µM, XE991 10 µM, ICA2743 10 µM, EtOH 0.1%), and the cells were kept in culture for a period of 4 days. The experiments were repeated 3 times.

### 4.5. Immunocytochemistry

The DRG explants, whether extracted from C57BL6/J or TSPO-KO mice under different conditions, were fixed with 4% paraformaldehyde (PFA) for 30 min at room temperature (RT). After 3 washing steps with 1X phosphate buffered saline (PBS), the coverslips were incubated in Antigen Retrieval Buffer (10 mM Sodium Citrate (Sigma Aldrich Chimie Saint-Quentin-Fallavier, France), 0.05% Tween20, pH 6.0) (preheated to 95 °C for 3 min. The samples were washed twice with PBS 1X, incubated for 30 min in a 0.1 M Glycine solution, permeabilized (0.25% Triton X100 (Sigma Aldrich Chimie Saint-Quentin-Fallavier, France), 0.1% Tween 20 in PBS 1X, 20 min at RT), and then blocked (2% BSA, 0.1% Tween 20, 10% Normal Donkey Serum (Sigma Aldrich Chimie Saint-Quentin-Fallavier, France), 1 h at RT). DRG explants were incubated overnight with a primary antibody against heavy neurofilaments (anti-NFH 1:600, Millipore (Molsheim, France) AB1989) at 4 °C. The next day, coverslips were washed three times with PBS 1X (10 min for every wash) and incubated with donkey anti-rabbit secondary antibody (1:1000, Jackson (Cambridge, UK) 711-545-152) for 1 h at RT in the dark. Samples were then washed with PBS 1X, and nuclei were stained with DAPI (1:1500, Thermo Fisher Scientific (Illkrich, France) 62247). Samples were then mounted on slides using a mounting medium (Inova Diagnostics Inc. (San Diego, CA, USA) 426950) and stored at 4 °C until confocal microscopy. 

### 4.6. Imaging and Image Analysis

Confocal imaging of fixed DRG explants was performed using the Leica SP8 microscope. Images were acquired as z-stacks at 10× magnification and then analyzed using ImageJ software version 1. For every experimental condition, 3–4 embryos from 2–3 gestating females were analyzed. Neurite length and density measurements were performed on z-projections with maximal intensity using the Neurite-J plugin of ImageJ software after thresholding the images. The plugin indicates the number of neurites at different distances from the explant [66]. Data were exported to Microsoft Excel, graphs were plotted using Prism v9.5.1 software, and areas under the curve (AUCs) were calculated based on the trapezoidal method.

### 4.7. Total RNA Isolation

RNA was extracted from the samples using the Ambion Life Technologies kit (cat. no. 15596018) with 500 µL of TRIzol reagent (Ambion Life Technologies (Saint-Aubin, France)) as described in the manufacturer’s protocol, with some modifications. In order to limit salt precipitation, isopropanol was used instead of 100% ethanol. Moreover, RNA was allowed to precipitate overnight in the presence of glycogen at −20 °C. The next day, RNA pellets were obtained after centrifugation, and possible contamination was eliminated after three washes with 70% ethanol. Tubes were kept open under the hood for 10 min to allow them to dry. RNA was resuspended in 20 µL of RNAse-free water containing 0.1 mM EDTA, pH 8.0 (Invitrogen (Waltham, MA, USA) AM9912). RNA samples were stored at −80 °C until RT-qPCR.

#### 4.7.1. RT-qPCR

Five hundred nanograms of total RNA was reverse-transcribed using the M-MLV RT buffer pack (Invitrogen, Charbonnières-les-Bains, France). After that, real-time PCR (qPCR) was conducted with the CFX96^TM^ Real-time system (Biorad, Hercules, CA, USA) using the Maxima SYBR Green Rox qPCR master mix (Thermo Scientific, Villebon-sur-Yvette, France), according to the manufacturer’s instructions. Each experiment was performed at least three times in duplicate. Relative gene expression of the targeted genes was determined by the 2^−ΔΔCt^ method and normalized to Gapdh mRNA expression levels. The expression levels of the treated DRG groups were normalized to those of the non-treated (NT) DRG.

#### 4.7.2. Primer Design and Efficiency

The TSPO, Kv7.2, Kv7.3, Stmn2, Peripherin, Nfh, and Gapdh primers used in this study were designed using the Oligo Explorer software version 1.2. Splice variants and the protein-coding sequence of the genes were identified using the National Center for Biotechnology (NCBI)’s database. For every gene, the common sequence between the different variants was identified using the Basic Local Alignment Search Tool (BLAST). Primer sequences were then generated by the Oligo Explorer software. Serial dilutions of cDNA, followed by qPCR, were used to calculate the amplification efficiency of the primers. The qPCR results were plotted as a standard curve against the respective concentrations of cDNA. Amplification efficiency was calculated by linear regression of standard curves using the following equation: E = 10^−(1/slope of the standard curve)^. Primer pairs with an efficiency between 95 and 100% as well as an R^2^ value (Determination coefficient) equal to or greater than 0.98 were chosen for this study (refer to Appendix A for the sequences of the chosen primers).

The other primers used in this study (Mpz, Mbp, Dhh, PLP, Krox20, CNPase, Cad19, and Tfap2α) were designed and chosen by our collaborators after the calculation of the amplification efficiency (E) by linear regression of standard curves using the following equation: E = 10^−(1/slope of the standard curve)^. Primer pairs that exhibited a theoretical E of 1.9–2.1 (95–105%) and an R^2^ value (Determination Coefficient) of 0.98 and above were chosen [39] (refer to Appendix A for the primer sequences).

### 4.8. Statistical Analysis

Statistics were computed using the GraphPad Prism 9.5.1 software. Differences between groups were calculated by one-way ANOVA followed by Tukey’s Multiple Comparison Test. A two-sided α = 0.05 was used as the significance level. Results are plotted as mean ± SD.

## Data Availability

The data presented in this study are available upon request from the corresponding author.

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
