# Peer review of "GRT-X Stimulates Dorsal Root Ganglia Axonal Growth in Culture via TSPO and Kv7.2/3 Potassium Channel Activation"

_ijms, 2024, doi:10.3390/ijms25137327_

Round 1

Reviewer 1 Report

Comments and Suggestions for Authors

The manuscript "The dual TSPO and Kv7.2/3 potassium channel activator GRT-X stimulates dorsal root ganglia axonal growth in culture” by El-Chemali et al., presents a good enough English. It has a good result to the field, but the manuscript needs major and minor revisions.

Major revision

1.     In the abstract they said that the mitochondrial translocator protein (TSPO) and Kv7.2/3 potassium channels have emerged as potential targets for enhancing nerve regeneration and alleviating neuropathic pain, but why they only determined both of them? They can obtain results such as Neurotrophic Factors Nerve Growth Factor (NGF): Promotes the survival, development, and function of neurons. Brain-Derived Neurotrophic Factor (BDNF): Supports the growth and differentiation of new neurons and synapses. Sodium Channels (Nav1.7, Nav1.8, Nav1.9): Voltage-gated sodium channels involved in pain signalling. Inhibitors can reduce pain by blocking abnormal electrical activity. Calcium Channels (Cav2.2): N-type calcium channels are involved in the release of neurotransmitters that transmit pain signals. Inhibition can reduce pain transmission. Transient Receptor Potential (TRP) Channels. G-Protein Coupled Receptors (GPCRs). Toll-Like Receptors (TLRs): Involved in the immune response and inflammation. Modulating TLRs can reduce neuroinflammation and pain. Cytokines and Chemokines: Targeting pro-inflammatory cytokines (e.g., TNF-α, IL-1β) can reduce inflammation and pain. Mitochondrial Function translocator Protein (TSPO): Involved in mitochondrial function and neuroinflammation. Ligands can promote nerve regeneration and reduce pain by modulating mitochondrial activity and reducing oxidative stress. Microglia and astrocytes activation. Extracellular Matrix Modulators Matrix Metalloproteinases (MMPs): Involved in remodelling the extracellular matrix, which is crucial for nerve regeneration. Inhibitors can prevent excessive tissue damage and support repair processes. Adenosine Receptors (A1, A2A). Stem Cells and Gene Therapy: Using stem cells to promote nerve regeneration and repair. This reviewer believes that the desired objectives have not been adequately addressed. It is a study limited to a few targets of what is intended to be analysed.

2. GRT-X. The beneficial actions on nerve viability and regeneration by GRT-X were attributed to its TSPO activation

3.     Author indicates: (Line 130) The experiment was repeated twice. The number of experiments should be increased. Two are not significant. This reviewer thinks that nothing can be concluded from Figure 3.

 4.     Data represent mean ± S.D over 3 independently performed manipulations or from 3 independently experiments. But for how many rats

 5.     Author indicated: For all genes studied, there were no differences between non-treated (NT) and the ethanol treated groups. Why, many authors have published that ethanol treated rats had differences compared with NT, with decreases in many genes studied in this manuscript.

 6.     Do you think administering an antiepileptic to a patient would be interesting? Perhaps the authors should test the toxic effects that this antiepileptic could produce.

Comments on the Quality of English Language

The english is more or less ok. 

Author Response

Comment 1: In the abstract they said that the mitochondrial translocator protein (TSPO) and Kv7.2/3 potassium channels have emerged as potential targets for enhancing nerve regeneration and alleviating neuropathic pain, but why they only determined both of them? They can obtain results such as Neurotrophic Factors Nerve Growth Factor (NGF): Promotes the survival, development, and function of neurons. Brain-Derived Neurotrophic Factor (BDNF): Supports the growth and differentiation of new neurons and synapses. Sodium Channels (Nav1.7, Nav1.8, Nav1.9): Voltage-gated sodium channels involved in pain signalling. Inhibitors can reduce pain by blocking abnormal electrical activity. Calcium Channels (Cav2.2): N-type calcium channels are involved in the release of neurotransmitters that transmit pain signals. Inhibition can reduce pain transmission. Transient Receptor Potential (TRP) Channels. G-Protein Coupled Receptors (GPCRs). Toll-Like Receptors (TLRs): Involved in the immune response and inflammation. Modulating TLRs can reduce neuroinflammation and pain. Cytokines and Chemokines: Targeting pro-inflammatory cytokines (e.g., TNF-α, IL-1β) can reduce inflammation and pain. Mitochondrial Function translocator Protein (TSPO): Involved in mitochondrial function and neuroinflammation. Ligands can promote nerve regeneration and reduce pain by modulating mitochondrial activity and reducing oxidative stress. Microglia and astrocytes activation. Extracellular Matrix Modulators Matrix Metalloproteinases (MMPs): Involved in remodelling the extracellular matrix, which is crucial for nerve regeneration. Inhibitors can prevent excessive tissue damage and support repair processes. Adenosine Receptors (A1, A2A). Stem Cells and Gene Therapy: Using stem cells to promote nerve regeneration and repair. This reviewer believes that the desired objectives have not been adequately addressed. It is a study limited to a few targets of what is intended to be analysed.

Response1: We agree that the main objective of the study was not adequately presented. The abstract has been rewritten to clarify the purpose of this study. Its objective was not to arbitrarily select 2 targets among the numerous ones that have been shown to play a role in axonal growth, but to study mechanisms of the powerful neuroactive compound GRT-X, which targets both TSPO and Kv7.2/3 channels. The minor issue 1 raised by Referee 2 was along the same line: as recommended, we placed more emphasis on GRT-X, now at the beginning of the title.

Comment 2: GRT-X. The beneficial actions on nerve viability and regeneration by GRT-X were attributed to its TSPO activation

Response 2: The beneficial effects of GRT-X on axonal growth were indeed completely abolished in DRG explants prepared from TSPO-/- mice. However, selective targeting of TSPO with XBD173 was only moderately efficacious and activating Kv7.2/3 channels with ICA27243 even tended to inhibit axonal elongation, confirming a previous report (Huang et al. 2016, Neuroscience 333, 356-367). This contrasts with the strong positive effect of GRT-X, which acts on both targets. These results provide additional support for the concept of a novel additive or even synergistic dual mode of action of GRT-X (Bloms-Funke et al 2022, Eur J Pharmacol 923, 174935). This is discussed lines 391-398. The model is presented in our scheme 1 (line 428).

Comment 3: Author indicates: (Line 130) The experiment was repeated twice. The number of experiments should be increased. Two are not significant. This reviewer thinks that nothing can be concluded from Figure 3.

Response 3: We apologize for the typing mistake. This experiment was performed four times and the information was directly corrected in the text (line 132 page 3).

Comment 4: Data represent mean ± S.D over 3 independently performed manipulations or from 3 independently experiments. But for how many rats ?

Response 4: We used 3 independent pregnant females mice for microscopy and for RT-qPCR experiments. Concerning microscopy, 3-4 DRGs were extracted from independent embryos (eg. 3-4 DRG from 3 to 4 embryos per mouse). For PCR experiments, all DRGs were used (please refer to figure 2).

Comment 5: Author indicated: For all genes studied, there were no differences between non-treated (NT) and the ethanol treated groups. Why, many authors have published that ethanol treated rats had differences compared with NT, with decreases in many genes studied in this manuscript.

Response 5: Indeed, ethanol is known to affect neural cells. However, we used ethanol at a very low concentration (0.1%) to avoid any potential toxicity and inhibitory effects on gene expression. As per this comment, we added a sentence in the results addressing this issue (line 213-216 page 7 and line 278-279 page 9).

Comment 6: Do you think administering an antiepileptic to a patient would be interesting? Perhaps the authors should test the toxic effects that this antiepileptic could produce.

Response 6: The scope of this study was limited to an ex-vivo model in which toxicity cannot be assessed. However, even at high doses, GRT-X has been shown to have no CNS side effects. Moreover, conditioned place preference studies showed no rewarding potential in rats, suggesting a lack of abuse potential (Bloms-Funke et al 2022, Eur J Pharmacol 923, 174935) (information added to the concluding paragraph in line 423).

We thank the reviewer for his valuable comments.

Reviewer 2 Report

Comments and Suggestions for Authors

The manuscript of El-Chemali et al. entitled: “The dual TSPO and Kv7.2/3 potassium channel activator GRT-X stimulates dorsal root ganglia axonal growth in culture” aimed to advance the understanding of the role of TSPO (the mitochondrial translocator protein) and Kv7.2/3 channels in nerve regeneration, as manifested by axonal elongation. Selected activators and inhibitors of TSPO and Kv7.2/3 channels were tested at the cellular as well mRNA levels using DRG explants extracted from wild type or TSPO-KO mice. The authors concluded that TSPO plays a key role in axonal growth. In addition, they suggested that the GRT-X could have additional intracellular targets besides TSPO and Kv7.2/3 channels. While reading the manuscript, I found some major and minor issues that need to be addressed.

Major issues

1. Figure 3, WT mice: Although the value of AUC normalized to NT (examined at DIV4) was not significantly different for the XE991 and ICA27243 groups compared to the NT, there was a noticeable decreasing trend. Interestingly, this decrease was statistically significant in DRG explants from TSPO-KO mice (Figure 7). It is intriguing that both the activation and inhibition of Kv7.2/3 channels resulted in similar effects on neurite growth. How can this be explained? Are XE991 and ICA27243 specific to Kv7.2/3 channels? By DIV8, this decreasing pattern was completely abolished when the AUC values for DRG explants from wild-type mice were compared. In this regard, the experimental dataset for TSPO-KO mice was not provided. If available, it would be appropriate to include it.

2. At DIV4, XBD173 (a TSPO activator) exhibited a stimulatory effect on axonal growth only when applied twice (Figs 3 and 5). In this case, the effect was similar to what was observed for GRT-X. However, mRNA data were provided only for single-dose experiments, during which differences between the GRT-X and XBD173 groups were observed in many tested proteins. These findings are not surprising, as unlike GRT-X, XBD173 did not produce any effect in single-dose experiments. It would be interesting to also present mRNA analysis for the two-dose experiments. Would changes in the tested proteins be similar for the GRT-X and XBD173 groups?

3. The Discussion section needs to be condensed. It should focus on explaining and evaluating what were found, showing how it relates to provided literature review, and making an argument in support of overall conclusion. It should not be a second results section.

Minor issues

1. The title of the manuscript should place more emphasis on GRT-X. It should appear at the beginning of the title.

2. In the Abstract: The authors mentioned that they applied agents such as XBD173, ICA-27243, and XE991 to the DRG culture (Line 24); however, they explained their reasoning only for XBD173. This could make it difficult for readers to understand the authors' intentions.

3. I recommend that the authors polish their English writing style to make the manuscript more attractive to a broader community. Some expressions were unusual and not commonly used in scientific literature. For example:

Line 32: “…..strongly suggesting the existence of additional targets and inviting to their exploration….”

Line 33:”….. These findings uncover a key role of TSPO in axonal elongation and in the dual target mode of action of GRT-X in DRG neurons…..”

Comments on the Quality of English Language

I recommend that the authors polish their English writing style to make the manuscript more attractive to a broader community. Some expressions were unusual and not commonly used in scientific literature. For example:

Line 32: “…..strongly suggesting the existence of additional targets and inviting to their exploration….”

Line 33:”….. These findings uncover a key role of TSPO in axonal elongation and in the dual target mode of action of GRT-X in DRG neurons…..”

Author Response

Comment 1: Figure 3, WT mice: Although the value of AUC normalized to NT (examined at DIV4) was not significantly different for the XE991 and ICA27243 groups compared to the NT, there was a noticeable decreasing trend. Interestingly, this decrease was statistically significant in DRG explants from TSPO-KO mice (Figure 7). It is intriguing that both the activation and inhibition of Kv7.2/3 channels resulted in similar effects on neurite growth. How can this be explained? Are XE991 and ICA27243 specific to Kv7.2/3 channels? By DIV8, this decreasing pattern was completely abolished when the AUC values for DRG explants from wild-type mice were compared. In this regard, the experimental dataset for TSPO-KO mice was not provided. If available, it would be appropriate to include it.

Response 1: Our results showing the inhibition of neurite growth by the Kv7.2/3 activator ICA27243 are concomitant with what is found in the literature (Zhou et al. (2016), Neuroscience 333, 356-367). However, the inhibition of Kv7.2/3 by XE991 showed the same effect. It is important to note that at this early embryonic stage (E13.5) in mouse DRGs, neurons exhibit key electrophysiological characteristics, including action potential generation and ion channel activity, but they continue to undergo significant maturation and refinement in their structure and function. Altering the electrophysiological properties of the neurons at this stage (by activating or inhibiting Kv7.2/3) may explain the similar results observed. We added this information to our discussion (line 313-317 page 11). In our study, the used of TSPO-KO model aimed to validate the dual GRT-X mode of action. That’s why we didn’t go further and conduct studies up to DIV4 in the TSPO-KO model.

Comment 2: At DIV4, XBD173 (a TSPO activator) exhibited a stimulatory effect on axonal growth only when applied twice (Figs 3 and 5). In this case, the effect was similar to what was observed for GRT-X. However, mRNA data were provided only for single-dose experiments, during which differences between the GRT-X and XBD173 groups were observed in many tested proteins. These findings are not surprising, as unlike GRT-X, XBD173 did not produce any effect in single-dose experiments. It would be interesting to also present mRNA analysis for the two-dose experiments. Would changes in the tested proteins be similar for the GRT-X and XBD173 groups?

Response 2: The scope of this study was to compare mRNA levels in wild type and TSPO-KO DRG at DIV4 after one treatment and to shed light on the fact that a single dose of GRT-X is sufficient to promote axonal elongation and to increase the expression levels of several neuronal and Schwann cell genes. We could expect the same outcome after  a 2 dose treatment. That’s why we didn’t go any further.

Comment 3: The Discussion section needs to be condensed. It should focus on explaining and evaluating what were found, showing how it relates to provided literature review, and making an argument in support of overall conclusion. It should not be a second results section.

Response 3: As recommended, we have shortened the discussion.

Comment 4. Minor issues

Comment 4.1: The title of the manuscript should place more emphasis on GRT-X. It should appear at the beginning of the title.

Response 4.1: This issue was addressed and the title was modified (line 1-2 page 1).

Comment 4.2: In the Abstract: The authors mentioned that they applied agents such as XBD173, ICA-27243, and XE991 to the DRG culture (Line 24); however, they explained their reasoning only for XBD173. This could make it difficult for readers to understand the authors' intentions.

Response 4.2: The abstract has been improved with focus on GRT-X.

Comment 4.3: I recommend that the authors polish their English writing style to make the manuscript more attractive to a broader community. Some expressions were unusual and not commonly used in scientific literature.

Comment 4.3.1: For example: Line 32: “…..strongly suggesting the existence of additional targets and inviting to their exploration….”

Response 4.3.1: This sentence was corrected (line 29 page 1).

Comment 4.3.2: Line 33:”….. These findings uncover a key role of TSPO in axonal elongation and in the dual target mode of action of GRT-X in DRG neurons…..”

Response 4.3.2: This sentence was corrected (line 32-35 page 1).

N.B. The manuscript has been carefully proofread.

We thank the reviewer for his valuable comments.

Round 2

Reviewer 2 Report

Comments and Suggestions for Authors

The authors responded appropriately to almost all issues. However, their explanation for why both inhibition and activation of Kv7.2/3 channels resulted in the same inhibitory effect on neuronal growth is very general. Both the activator and inhibitor were added at the same time (DIV1), thus, the neuron maturation state at the time of addition was the same. I agree that this interesting phenomenon is related to neuronal excitability because Kv7.2/3 channels serve to regulate the action potential threshold. In summary, the provided explanation could be accepted, as at this stage, other explanations would be highly speculative.

Author Response

Comment 1: The authors responded appropriately to almost all issues. However, their explanation for why both inhibition and activation of Kv7.2/3 channels resulted in the same inhibitory effect on neuronal growth is very general. Both the activator and inhibitor were added at the same time (DIV1), thus, the neuron maturation state at the time of addition was the same. I agree that this interesting phenomenon is related to neuronal excitability because Kv7.2/3 channels serve to regulate the action potential threshold. In summary, the provided explanation could be accepted, as at this stage, other explanations would be highly speculative.

Response 1: 

We thank the reviewer for his comment. We agree that any other explanation would be highly speculative at this stage. As we do not have a strong argument to explain this phenomenon, we preferred to limit our explanation.